# Modern Approaches to Acellular Therapy in Bone and Dental Regeneration

**DOI:** 10.3390/ijms222413454

**Published:** 2021-12-15

**Authors:** Alexey A. Ivanov, Alla V. Kuznetsova, Olga P. Popova, Tamara I. Danilova, Oleg O. Yanushevich

**Affiliations:** 1Laboratory of Molecular and Cellular Pathology, A.I. Evdokimov Moscow State University of Medicine and Dentistry, 20 Delegatskaya Str., 127473 Moscow, Russia; avkuzn@list.ru (A.V.K.); petrovnapopova@rambler.ru (O.P.P.); danilova.tam@yandex.ru (T.I.D.); 2Koltzov Institute of Developmental Biology, Russian Academy of Sciences, 26 Vavilov Str., 119334 Moscow, Russia; 3Department of Paradontology, A.I. Evdokimov Moscow State University of Medicine and Dentistry, 20 Delegatskaya Str., 127473 Moscow, Russia; olegyanushevich@me.com

**Keywords:** endogenous regenerative medicine, bone regeneration, dental regeneration, decellularized extracellular matrix, growth factors, extracellular vesicles, miRNAs

## Abstract

An approach called cell-free therapy has rapidly developed in regenerative medicine over the past decade. Understanding the molecular mechanisms and signaling pathways involved in the internal potential of tissue repair inspires the development of new strategies aimed at controlling and enhancing these processes during regeneration. The use of stem cell mobilization, or homing for regeneration based on endogenous healing mechanisms, prompted a new concept in regenerative medicine: endogenous regenerative medicine. The application of cell-free therapeutic agents leading to the recruitment/homing of endogenous stem cells has advantages in overcoming the limitations and risks associated with cell therapy. In this review, we discuss the potential of cell-free products such as the decellularized extracellular matrix, growth factors, extracellular vesicles and miRNAs in endogenous bone and dental regeneration.

## 1. Introduction

Regenerative medicine, and, in particular, tissue engineering, are considered to be promising strategies for the repair of lost/damaged tissues, and aims to improve a patient’s quality of life. Approaches to tissue reconstruction are based on the use of stem cells (SCs), growth factors, signaling molecules, scaffolds and gene therapy. Stem cells, primarily mesenchymal SCs (MSCs) or progenitor cells, obtained from various tissues are attractive therapeutic agents. Their advantages are not only their rapid buildup in required amounts but also differentiation into various cell types to enable the modeling of various technologies for the reconstruction of lost or damaged tissues and organs [1,2]. Compared to regenerative potential, the SC capability of immunomodulation plays a no less important role in achieving a successful result [3]. To date, the use of SCs is considered to be the main strategy in regenerative medicine, including restorative dentistry. Based on therapeutic agents with SC expansion ex vivo, the reconstruction of the lost structures of various tissues was demonstrated in numerous preclinical and clinical studies [4]. However, tissue regeneration by SC transplantation is hindered by many factors, including immune repulsion, pathogen transfer, oncogenesis, the accumulation of genomic alterations and age-related genetic instability, problems with ex vivo manipulations with cells, time-consuming procedures, high costs and difficulties in obtaining regulatory approval [5]. 

An approach known as cell-free therapy has rapidly developed in regenerative medicine in the past decade, due to the growing volume of knowledge of the SC mechanisms of action. Together with an understanding of the paracrine effects of exogenously administered SCs, the molecular mechanisms and signaling pathways involved in the intrinsic potential of tissue regeneration started to be understood. This prompted the development of novel strategies to control and intensify these processes during regeneration. The use of SC mobilization/homing technology for regeneration based on endogenous healing mechanisms has become a new concept in regenerative medicine and is called endogenous regenerative technology (ERT) [6], endogenous regenerative medicine (ERM) [7,8] or autotherapies [9]. ERM/ERT is especially promising in restorative dentistry due to the large number of patients and one of its most remarkable advantages; the regeneration of merely a small amount of tissue can be very efficient for a patient [10]. Approaches using chemoattractant gradients to monitor tissue regeneration without ex vivo cultured cells are preferable to treatment techniques based on transplanted autologous or allogeneic SCs with limited potential for clinical use. 

The development of regenerative approaches in ERM/ERT requires, aside from the knowledge of biological SC homing–regulating signals, requires a comprehensive idea of the characteristics of the resident SCs’ microenvironment/niche, as well as of the extracellular factors involved in SC self-maintenance, to manipulate these cells. The key function of SC niches is to maintain a constant number of slowly dividing cells to balance the proportion of quiescent and activated cells. In a niche, as it is known, SCs are under the spatio-temporal control of an enormous number of factors, including chemokines, cytokines, growth factors, ligands, insoluble transmembrane receptors, proteases, adhesion molecules (selectins and integrins) and extracellular matrix (ECM) molecules [7]. Additionally, external tensile, compressive and shear forces have a massive effect on the phenotype of cells, the properties of the ECM, and the general functions of the niche. In turn, cells in tissues affect their microenvironment by internal mechanical forces—adhesion interactions of their cytoskeleton with the ECM and adjacent cells via the niche [11]. Thus, the interaction and optimization of every niche component involved in ERM is especially important for understanding how the required cell response should be made safe and efficient for therapy [12].

An ever-increasing amount of currently emerging data indicates that the use of cell-free therapeutics leading to endogenous SC recruitment/homing has advantages for overcoming restrictions and risks associated with the use of cell-free therapy, including tumorigenesis, unwanted immune responses and transfer of pathogens. Additionally, it has significant advantages in production, storage and standardization [13,14,15]. Thus, the use of cell-free products in regenerative medicine can improve migration, proliferation, differentiation and metabolism of various resident SCs, which provide for the regulation of their spatially correct arrangement and will stimulate the endogenous regeneration of damaged tissues [16]. 

This review deals with the potential of such cell-free products as decellularized ECM (dECM), growth factors, extracellular vesicles and miRNAs in bone and dental regeneration. We discuss the important roles of these cell-free products in forming a favorable niche for resident SC homing. 

## 2. Extracellular Matrix

As we mentioned above, the leading strategy in ERM is the use of various factors (Figure 1) that stimulate recovery mechanisms by recruiting endogenous SCs into injured areas [17]. 

It is known that, as well as growth factors and various signaling molecules, the traffic of SCs, their oriented migration, survival, self-renewal and differentiation can be regulated and enhanced by the ECM [18]. 

ECMs are highly specialized and dynamic three-dimensional frameworks that aid the adhesion and functioning of various cells that form the basis of tissues. ECMs consist of numerous fibrillar components such as collagens, fibronectin and elastin, and nonfibrillar molecules such as proteoglycans, hyaluronan and glycoproteins, including matrix cell proteins. They interact with one another via numerous receptors, including integrins, discoidin domain receptors (DDR), proteoglycan surface receptors and hyaluronan receptors such as CD44, RHAMM, LYVE-1 and layilin, creating a multicomponent structural network [19]. For instance, collagen, vitronectin and laminin are common partners for binding integrins. Some integrins can also bind to intercellular adhesion molecule 1 (ICAM1) and vascular cell adhesion molecule 1 (VCAM1), which are part of the SC microenvironment [20]. In addition to classical receptors, ECM molecules also interact with and regulate signal transfer via other non-traditional receptors, including growth factor receptors and Toll-like receptors (TLRs) [19]. It is known that the differential expressions of certain receptors determine the type of niche into which cells migrate. Thus, for instance, integrins β1, α5 and αV are usually expressed into adult SCs [20]. ECMs regulate the proliferation, survival, migration and differentiation of cells via matrix–cell interactions. Thus, ECM molecules interact with surface receptors of various cell types, including fibroblasts, immune cells, endothelial cells, epithelial cells and pericytes, by regulating the phenotypes and functions of these cells for tissue homeostasis maintenance. 

The ECM molecular composition and structure differ in different tissues and change noticeably at the reconstitution of normal tissue, as well as in the progression of various diseases. 

ECMs in tissues such as cartilage and bone differ in their composition and structure from ECM in connective tissue, as they bear significant mechanical loads. The bone contains a specialized ECM, which essentially consists of collagen I, III and V fibrils (Table 1). Collagen I is a predominant protein that serves as a site for the nucleation of hydroxyapatite and the deposition of crystals on its fibrils. The main ECM components are synthesized by osteoblasts, although terminally differentiated osteoblasts called osteocytes also produce matrix components, such as small integrin-binding ligand N-linked glycoproteins (SIBLING): dentin matrix acidic phosphoprotein 1 (DMP1) and matrix extracellular phospho-glycoprotein (MEPE), involved in phosphate metabolism and bone mineralization [21]. Osteocalcin, osteopontin/bone sialoprotein, as well as small lecithin-rich proteoglycans, such as keratocan and asporin, are also involved in bone mineralization. Decorin, biglycan, asporin, osteonectin/secreted protein acidic and rich in cysteine (SPARC), and thrombospondin are engaged in collagen fibrillogenesis and/or bioavailability/transmission of signals from transforming growth factor (TGF)-β. The mineralized ECM imparts the tissue with rigidity and mechanical strength, and all its components contribute to correct tissue functioning [22]. The ECM acts not only as a physical framework for cells and as a store of signaling molecules, but also as the main regulator of the behavior of SCs in a niche [23,24]. For this reason, various secretory SC products as well as ECM products and ECM-based bioscaffolds are considered, at present, as a new class of biopharmaceuticals in regenerative medicine [25]. As compared with cell-based therapy, the use of biomaterials is simpler and sufficiently reliable for maintaining high levels of endogenous tissue regeneration. Therefore, a well-designed biomaterial-based niche has the potential for activating and recruiting a sufficient amount of SCs from adjacent tissues for safe, functional regeneration [8,26].

In view of the organization of various organs as three-dimensional structures, it is essential to choose for their regeneration an underlying scaffold mimicking the ECM in native tissue. The choice of scaffold materials and design impacts the therapeutic potential and the number and invasiveness of associated clinical procedures [12]. As scaffolds may change their physical and chemical properties and transfer mechanical forces in vivo in response to various internal and external stimuli, they may contribute both to regeneration and to the development of a reaction to a foreign body and fibrosis [11]. This necessitates the complete understanding of cell–scaffold interactions. To date, these interactions are known to be mediated by various adhesion molecules, including integrins and cadherins, which are of crucial importance for cell migration and localization [27]. It is noteworthy that the successful penetration of cells and their presence in the scaffold is regulated by biomaterials’ surface features and cell–matrix interactions of cells with biomaterial [28,29]. Thus, the scaffold constructions designed by tissue engineering can ensure a suitable microenvironment for resident SCs’ homing and the controlled release of biological signals, including matrix-associated growth factors (fibroblast growth factor (FGF), TGF-β, bone morpho-genetic protein (BMP)) [30,31], which aid with model physiological processes, including tissue morphogenesis and regeneration [11].

Overall, scaffolds should meet four main criteria with respect to (1) their shape (correspondence to the geometry of complex three-dimensional defects); (2) function (temporary maintenance of the functional and biomechanical conditions in the course of healing); (3) formation (contribution to regeneration); and (4) fixation (light interaction and integration with adjacent tissues) [32].

Biomimetic frameworks actively developed at present in tissue engineering are formed based on purified ECM components and synthetic or natural polymers [33]. Biomimetic designs attract the ever-increasing attention of many researchers in biomaterials, regenerative biology and regenerative medicine communities. Unfortunately, as of now, bioengineers have failed to construct the basic elements of native tissue [5,34]. However, scaffolds consisting of the main ECM components and/or structures can mimic an in vivo microenvironment occurring in the course of tissue regeneration and, therefore, can contribute to endogenous tissue reconstitution. For instance, a hierarchically structured nanohybrid framework containing a bone-like nano-hydroxyapatite (n-HA) contributes to a homogeneous distribution of n-HA after in vivo transplantation and to interfacial interactions by recruiting endogenous cells for successful bone regeneration in situ [35]. The possibility of constructing a biological activity and changing the parameters of biomaterials’ properties significantly increases the number of potential applications and improves biomaterials’ characteristics in vivo. Recent advances in the biotechnologies of the development of multiphase scaffolds, such as electrospinning and 3D bioprinting, enable the high-accuracy formation of a complex architecture comparable with native bone architecture both in the shape and structure [5]. For instance, Kankala et al. demonstrated a 3D porous scaffold using the innovative combinatorial 3D printing and freeze-drying technologies on gelatin (Gel), n-HA and poly(lactide-co-glycolide) (PLGA) for bone regeneration [36].

Along with biomimetic frameworks, a strategy, which aims to use decellularized matrices that possess the advantage of a great similarity with the tissue to be replaced, is also being actively developed in tissue engineering. It is crucial to choose proper decellularization methods for obtaining decellularized matrix biomaterials [37], which will greatly affect the ultrastructure, composition and biological actions. It is commonly acknowledged that it is essential to remove cellular elements such as the cell membrane, nucleic acids and mitochondria as much as possible, but keep the functional compositions. Currently, there are many conventional methods to prepare decellularized matrix-based scaffolds. This can be accomplished using physicochemical and chemical methods, including freeze–thaw methods, ultrasonication and freeze drying, treatment with chemical detergents such as Triton X-100 and SDS, or enzymatic treatment with DNase and RNase. Triton X-100 is better for preserving the ECM architecture compared to freeze–thaw cycles [38]. The key part of the decellularization assessment is the analysis of changes produced in the dECM. Providing the presence of most ECM components after the decellularization is of key significance for maintaining its functionality. As ECM-based scaffolds produced by mammalian tissue decellularization exhibit no immune responses, and by their nature contain tissue-specific and matrix-associated factors involved in cell growth and differentiation, they are actively used in bone and dentistry regeneration [39]. In vitro, decellularized bone ECM enhances the osteogenic differentiation of rat MSCs [40], human embryonic SCs [41] and human adipose-derived SCs [42]. In vivo, dECM displayed efficient engraftment and vascularization and was able to undergo remodeling onto an immature osteoid tissue [43]. The dECMs can efficiently integrate into the defect zone and promote bone repair [44]; decellularized periodontal ligaments can reconstruct periodontal tissues by recruiting host cells and evoking their correct orientation in the ligament, which can serve as a novel approach to periodontal treatment [45]. The dECM implanted in the mouse calvarial defect model improved not only new bone formation without any further inflammatory reaction, but also the density of these formations [46]. 

In general, decellularization techniques preserve the capability of bone ECM scaffolds to induce the osteogenic differentiation of cells in vitro and to promote angiogenesis and cell infiltration in vivo [40,47]. Moreover, dECMs that preserve native tissue structural components and contain many diverse biological signals and growth factors can control the homing and differentiation of endogenous SCs [48,49,50]. For this reason, decellularization can be used for the production of bioscaffolds that preserve biomechanical properties and maintain the complex three-dimensional structure of native tissues, thereby providing for successful tissue regeneration. The past decade saw a rapid development of dECM biomaterials, which proved to have numerous benefits and potential in both preclinical and clinical applications [51]. The cell-derived ECM is one of the modifications of dECM [52]. The widespread use of this scaffold in the future will rely on moderating the cost of generating ECM from cells and could present a new avenue in bone regeneration, since they could be engineered to produce scaffolds with controllable biological effects. ECM coating could be a promising concept for the development of biologically active biomaterials that systematically affect cell behavior. Thus, the use of bioscaffolds to create a “niche” for successful cell recruitment and survival [53] is an attractive strategy of crucial significance in the regeneration of complex anatomical structures [30,31]. However, the existing challenges are still hindering the more profound applications of dECM biomaterials, for example, the issue on how to retain the active ingredients and structure in the initial preparation, as well as the subsequent processing, sterilization, preservation, and other processes. A more detailed mechanism for the interaction of extracellular matrices with cells and in vivo microenvironments is currently unclear. In addition, the specific composition of a decellularized scaffold that promotes cell behavior, tissue regeneration and angiogenesis is still unclear, and related cellular and molecular mechanisms are also worth studying. All of these obstacles limit further possibilities for the use of more advanced applications in the clinic. Given the exquisite complexity of regenerative mechanisms, multiprong bioengineering approaches are needed to enable spaciotemporal control of SC recruitment.

## 3. Growth Factors and Signaling Molecules

The approach using only scaffolds is often insufficient for reconstructing a biologically suitable extracellular SC microenvironment/niche. In this case, combinations of ECM molecules and growth factors are used [54]. Growth factors and signaling molecules can stimulate chemotaxis, proliferation, differentiation, ECM synthesis and angiogenesis. The biological functions of these molecular mediators widely vary, but their choice as candidates for regenerative therapy of bone and dental tissues is based on their important role in the development of these tissues and their healing [30] (Figure 2). Thus, for instance, during the early phase of bone healing, platelet activation and subsequent degranulation provide a burst of cytokines directly at the injury site. These factors cause the migration of innate immune cells to the site of damage. Recruited immune cells secrete paracrine factors (e.g., TGF, BMP, FGF, vascular endothelial growth factor (VEGF), focal adhesion kinases) that form a favorable microenvironment at the site of damage. This microenvironment promotes the migration of both circulating and local resident populations of reparative cells, such as SCs and progenitor cells, by means of complex signal cascades [7]. As a result, the recruited MSCs begin to differentiate into fibroblasts, chondroblasts, and osteoblasts. BMP-2 and BMP-7 play an important role in the induction of MSC differentiation into osteoblasts [5]. Additionally, TGF-β/BMP, by interacting with various pathways—Wnt, MAPK, Notch, Hh, Akt/mTOR and miRNAs—activates BMP-stimulated signal transmission and induces endogenous bone regeneration [55].

Promising results of preclinical and clinical research led to the subsequent introduction of various growth factors to the commercial market for regeneration of soft and hard tissues [56]. However, there are several known problems associated with growth-factor-based therapy, which should be strictly taken into account: short periods of half-decay in vivo, side effects due to the introduction of several or high doses to achieve efficient therapy, unidentified key growth factors for particular tissues and the possible denaturation of protein during manipulation [5]. The use of scaffolds with required biomolecules adsorbed on them can avoid these problems in the induction of the endogenous regeneration of bone and dental tissues. For instance, the bone ECM can be used as a scaffold for recruiting endogenous progenitors using various signaling molecules or angiogenic factors such as VEGF, proinflammatory cytokines such as tumor necrosis factor TNF-α and interleukin-1, as well as BMP for the stimulation of bone regeneration [54]. Several in vitro and in vivo studies have shown that the addition of various signaling molecules and growth factors, such as granulocyte colony-stimulating factor, stromal cell-derived factor (SDF), basic FGF (bFGF) and VEGF, to different (natural and syn-thetic) scaffolds enhances the regeneration of intracanal pulp-like tissues due to the stimulation of dentin formation, mineralization, neovascularization and innervation [57]. Decellularized dentin can be modified by platelet-rich fibrin (PRF) to provide signals for MSC recruitment from the circulation and the periodontal ligament for the regeneration of cementum and tissue similar to the periodontal ligament with oriented fibers, which ultimately restores the interface between soft and hard tissues [58]. 

The combination of recombinant human BMP-2 (rhBMP-2) on an absorbable collagen sponge carrier was shown to induce bone formation in a number of preclinical and clinical investigations [59]. The main issue associated with the absorbable collagen sponge is an initial burst release of rhBMP-2 into the local environment, leading to heterotrophic ossification [60]. One reason that BMP carriers are loaded with supraphysiological concentrations is likely related to the need to overcome the regulating factors of BMP inhibitors in order to achieve a therapeutic response. These inhibitors are present within the BMP signaling cascade at intracellular locations, as pseudo-receptors, and in extracellular locations [61]. In order to successfully decrease the therapeutic concentration of BMPs, novel carrier systems that maintain or enhance rhBMP-2 bioactivity must be designed and the negative feedback signaling caused by BMP antagonists must be addressed.

A particularly intriguing approach is the modification of known growth factors with so-called superaffinity domains, which allows these growth factors to achieve effects at a lower effective dose through better binding affinity to their carrier material or ECM proteins [62]. 

Thus, the main tendencies in acellular bone tissue engineering today are directed to the creation of an ECM-based bioscaffold, usually by including several key growth factors for mimicking the natural bone structure and developing an environment for maintaining osteogenesis, osteoconduction and/or osteoinduction [63]. Although the use of growth factors appears to be an extremely attractive strategy for establishing a microenvironment around an implanted scaffold, problems associated with its efficiency and safety remain. The targeted delivery of growth factors can be a complicated problem, because they rapidly degrade and diffuse into surrounding tissues. The potency of each cytokine in a cocktail differs from its individual action; thus, the synergistic or antagonistic effects of multiple cytokines on a given cell population must be tested under different in vivo situations [64]. Further research into the molecular pathways underlying the process of SC recruitment is required to assess the real potential of the growth factors. Studies that aim to trace in vivo SC migration in response to the local gradients of the growth factors may help find new biological signals and improve the selectivity of the existing signals [17].

## 4. Secretome and Extracellular Vesicles

As noted above, cell-free therapy is an approach in regenerative medicine that makes use of SCs and progenitor cells as a source of therapeutic molecules, but not as therapeutic agents. SCs secrete various factors called secretomes. These factors can be found in a medium where SCs are cultured, i.e., in a conditioned medium. A conditioned MSC medium for cell-free therapy was recently considered to be a source of various factors. As some studies showed, the secretome can, without SCs, cause a restoration of tissues/organs during damage [65,66,67], including when used in restorative dentistry [68,69,70,71,72,73,74]. The advantages of this cell-free approach are determined by the paracrine effects of biological molecules on damaged organs and tissues in the absence of the long-term engraftment and survival of transplanted cells. An additional advantage of using secretomes, preselected and screened by a variety of parameters (including exogeneous pathogens, donor age, multipotency, profile, quality control), is the ability of their rapid use for patient treatment without the isolation of SCs and their subsequent cultivation [14]. Moreover, secretomes can increase stability with reduced requirements for storage conditions in frozen form [75]; they can be used immediately upon thawing, and can be freeze-dried to yield a finished product while maintaining functionality [76]. All of these advantages of secretomes, as compared with cell therapies, can reduce production costs [77]. Additionally, MSC secretomes from tissues of various origins can be enclosed into diverse biomaterials. The potential of such combinations was shown in a number of preclinical studies during the healing of defects of periodontal tissue [72], alveolar bone [69], mandibular angles [70], calvarial bone [68,78,79] and maxillary sinus floor elevation [71]. The efficiency of such combinations was also demonstrated for bone formation in age-related osteoporosis [76]. Thus, for instance, a conditioned medium produced from cultured periodontal ligament SCs enhanced the periodontal regeneration in a rat periodontal defect model in a concentration-dependent manner by suppressing the inflammatory response and decreasing the levels of mRNA and TNF-α [74]. The literature also reports a clinical application of an MSC-conditioned medium to a limited number of patients diagnosed as needing bone augmentation before dental implant placement [73]. In this case, the secretome contained several growth factors in relatively small amounts, such as insulin-like growth factor (IGF)-1, VEGF, TGF-β1 and hepatocyte growth factor. No systemic or local complications were observed throughout the study. An X-ray evaluation confirmed by histological examination revealed early bone formation in all cases. In addition, the infiltration of inflammatory cells was insignificant in all histological specimens [73]. 

Although growth factors and cytokines are a significant part of the SC secretome, these cells also produce extracellular vesicles (microvesicles, exosomes, apoptotic bodies, microparticles), which represent nanosized vesicles enclosed in a lipid membrane. Extracellular vesicles are present in all body fluids and are secreted by all types of cells in the human body. They are classified by biogenesis and size [80,81]. The following types of vesicles are distinguished: exosomes, smaller than other vesicles (approximately 40–100 nm), released from the cell via the multivesicular endosomal pathway; microvesicles (about 20–1000 nm), released by budding-off from plasma membrane segments; apoptotic bodies (1000–5000 nm), which are formed by the fragmentation of dying cells and can contain various cell parts; and microparticles (50–80 nm), the less investigated subgroup of extracellular vesicles [80,81]. Of interest in tissue engineering, among extracellular vesicles secreted by MSCs, are exosomes [65,66] and microvesicles [14] due to their unique ability to transfer lipids, proteins and various RNA forms (including miRNAs) into adjacent cells to mediate a broad range of biological functions [14,15,82]. The most widespread and currently used technique for the isolation of extracellular vesicles is differential centrifugation, where microvesicles are isolated by ultracentrifugation at 10,000–20,000× *g* [80,81]. This approach requires a large expenditure of time and an ultracentrifuge; additionally, the yield of extracellular vesicles is comparatively small, and the purity of produced samples can be low [80]. Purer fractions of extracellular vesicles are obtained by density gradient centrifugation, exclusion chromatography, polymer-based sedimentation and immunoaffinity [80,81]. The main exosome markers are tetraspanins, CD63 and CD9; apoptosis-linked gene-2 interacting protein X (ALIX); and tumor susceptibility gene 101 (TSG101), and the microvesicle markers are CD40 and CD62 [80]. An analysis of the proteome of extracellular vesicles originating from MSCs revealed 730 various proteins. They include surface receptors and signaling molecules involved in the self-renewal and differentiation of MSCs, as well as in the proliferation, adhesion and morphogenesis of many other types of cells [83]. Given the ability of extracellular vesicles to transfer bioactive components and to overcome biological barriers, exosomes and microvesicles are increasingly more often studied as potential therapeutic agents, as well as by means of their delivery [81]. To date, the therapeutic potential of extracellular vesicles in the absence of MSCs was shown in various disease models [84,85]. In addition, it was reported that extracellular vesicles are capable of exhibiting functional properties, similar to those of cells from which they are derived, and have no clear side effects such as immunogenicity or oncogenicity [86,87,88]. In connection with the prospects of using extracellular vesicles in various diseases, approaches are being developed to increase their therapeutic efficiency [80]. For instance, the overexpression of miR-140-5p in exosomes obtained from human synovial MSCs enhances the regeneration of cartilage tissue and prevents osteoarthritis of the knee joint in a rat model [89]. In addition to the modification of exosomes, to stimulate the release of soluble factors or extracellular vesicles from MSCs with strong proangiogenic and anti-inflammatory properties that promote survival, the cells themselves are subjected to genetic modification [77]. In addition to the above, for certain indications, it is important to choose the tissue source from which the MSCs secretome is obtained, since there is a significant difference in the levels of growth factors and extracellular vesicles secreted by the MSCs, depending on the tissue of their origin; extracellular vesicles, in turn, are enriched with various types of regulatory RNAs and proteins. For example, for immunomodulatory tasks, it is preferable to use the secretome of umbilical cord MSCs, and for bone regeneration, the secretome of bone marrow MSCs [77]. Moreover, it is important not only to choose the source of tissue from which the MSC secretome is obtained; the origin of the tissue (adult or embryonic) is also important, since this can affect both the production of extracellular vesicles and their content [90]. It is also necessary to consider the fact that miRNAs in extracellular vesicles do not just reflect cellular contents. Some miRNAs are present both in MSCs and in their respective extracellular vesicles, while others are represented selectively [77,90].

Mechanisms of tissue repair in vivo mediated by extracellular vesicles originating from MSCs include immune modulation, enhanced angiogenesis, the inhibition of apoptosis and the reduction in fibrosis [77]. Therefore, extracellular vesicles, as well as secretomes obtained from MSCs, can play an important role in tissue regeneration and represent a potential alternative to SC therapies [81]. 

The delivery routes of extracellular vesicles into the body include intravenous, intraperitoneal and subcutaneous injections. Extracellular vesicles can also be injected directly into the site of injury. Local delivery, as opposed to systemic delivery, can help ensure that any effects of bioactive factors are not widespread [77]. Extracellular vesicles introduced by bolus injections are rapidly sequestered and excreted, which requires repeated injections during the healing period. The short-term retention of extracellular vesicles after in vivo delivery is recognized as a major obstacle for their clinical use, since they are rapidly excreted after systemic delivery by innate immunity mechanisms [91]. These problems echo those observed with growth factor therapies. In order to overcome the rapid rate of excretion and localize the activity of extracellular vesicles, biomaterials for their delivery are used. It was shown that the incorporation of secretomes into a biomaterial matrix increases their bioavailability after delivery, ensures their stable and controlled release, maintains their stability and potentially increases therapeutic efficacy [77,92,93]. For instance, there are reports of the use of exosomes obtained from MSCs of various origins, including in combination with bioactive materials for the formation of bone tissue on models of calvarial bone defects [94,95,96], femoral fracture [92] and bone and cartilage defects [97,98]. Thus, it was shown that the use of exosomes from MSCs obtained from induced pluripotent cells, together with tricalcium phosphate, can significantly contribute to osteogenesis in a model of critical-size calvarial bone defects in rats. Among the mechanisms by which combined frameworks stimulate osteogenesis, researchers suggest the involvement of exosomes in the activation of the PI3K/Akt signaling pathway [94]. Another study found that miR-196a plays an important role in the regulation of osteoblast differentiation and the expression of osteogenic genes [95]. 

To date, the clinical evaluation of secretomes, and particularly those of extracellular vesicles obtained from MSCs, is limited, but significant trials are needed to establish the effectiveness of this approach. Several clinical trials and case studies explored the potential of conditioned media and reported the safety and potential efficacy. Leveraging the knowledge base established by the successes and challenges in manufacturing cells, the MSC-derived extracellular vesicle field is well-poised for a quick translation from research to the clinic [99,100]. However, before the clinical translation of extracellular vesicles derived from MSCs and soluble factors, many obstacles should be overcome, including the determination of the optimal tissue source of MSCs, dosages and routes of administration, the understanding of bioactive components and mechanisms of action, and achievement of the scalability and GMP-grade products [77]. 

## 5. Noncoding RNAs

As mentioned above, extracellular vesicles, aside from proteins and lipids, contain regulatory noncoding RNAs. In many preclinical studies, therapeutic effects mediated by extracellular vesicles are associated with the contents of their nucleic acids. Interestingly, the treatment of extracellular vesicles with RNase prevented their effect on kidney recovery in a mouse model of acute kidney injury, indicating the presence of RNAs as the main therapeutic component [101]. 

Among regulatory noncoding RNAs, small interfering RNAs (siRNAs) and microRNAs (miRNAs) are distinguished. siRNAs and miRNAs have much in common, but their mechanisms of action and clinical applications are different [102]. siRNAs and miRNAs are largely similar in their physico-chemical properties; both are short duplex RNA molecules about 20–22 nucleotides long that suppress the activity of target genes at the posttranscriptional level. The main difference between siRNAs and miRNAs is that the former are highly specific to only one mRNA, while the latter have multiple targets [102]. siRNAs and miRNAs are extensively used to diagnose and treat various diseases at both the cellular and molecular levels. However, the application of therapies with these nucleic acids for bone regeneration has not progressed to clinical trials. One of the main challenges for siRNA and miRNA therapies is the lack of effective and safe delivery vehicles that can provide the sustained release of RNA molecules at the target site of bone defects and in surrounding cells [103]. The delivery of siRNA or miRNA alone is not successful due to the susceptibility of these RNA molecules to degradation and the overall negative charge that prevents them from passing through the cell membrane [104]. 

To target cells, siRNAs and miRNAs are delivered by exosomes; into exosomes, they can be successfully loaded by electroporation [105]. In addition, biopolymer hydrogels can be used as siRNA and miRNA delivery vehicles [106]. Biomaterial-augmented miRNA and siRNA delivery approaches show promise for achieving the robust and precise control of gene expression [107]. For instance, the use of chitosan hydrogel loaded with RANK-specific siRNA has shown a successful downregulation of osteoclast activity in vitro [108]. siRNA against noggin, which is an antagonist to the activity of BMP-2, -4, -5, -6 and -7, was delivered from a synthetic polymer and successfully enhanced the osteogenic activity of MSCs in vitro [109].

The most studied class of RNAs enclosed in extracellular vesicles are miRNAs. The complete complementarity of sequences between miRNA and its potential target, mRNA, is not required to suppress gene expression, and a match of only six base pairs is sufficient [102]. Consequently, one miRNA has the potential to simultaneously control the translation of hundreds of genes, which often work together as a network within the same signaling pathway or biological process, including cellular differentiation, proliferation, apoptosis and inflammatory reactions [102,110]. Furthermore, since miRNAs remain stable in extracellular fluids due to their packaging, they are ideally suited for use as noninvasive biomarkers of various diseases. Thus, for instance, many reports show the differential expression of miRNAs between healthy and periodontitis gingival tissues [111,112,113,114]. The change in miRNA expression in gingival tissues is reflected in biological fluids, such as serum, saliva and crevicular fluid of the gingiva, which can serve in the diagnosis of periodontitis [115]. However, in order to use miRNAs in diagnostics, standardized criteria and protocols of pre-analytics, measurements and analysis must be established to obtain comparable results in different studies [116]. 

In recent years, the role of miRNAs in the posttranscriptional regulation of genes has come to the fore, with convincing evidence indicating the important role of miRNAs in regulating a wide range of fundamental biological processes, including the regulation of maintenance and differentiation of MSCs and/or other progenitor cells, as well as mechanisms of endogenous tissue repair [117]. The regulation of the above processes may be important for the application of various therapeutic strategies in the regenerative medicine of tissues or for a better understanding of the molecular mechanisms of human diseases associated with SC differentiation. A number of studies showed the differential expression of miRNAs when MSCs switch from a proliferative state to a differentiated one [118] or during the differentiation of MSCs in alternative directions, namely in osteogenic [119], adipogenic [120,121] or chondrogenic ones [122]. For instance, the overexpression of miR-155 in MSCs during osteogenic differentiation leads to a decrease in alkaline phosphatase and alizarin red S staining, as well as a decrease in the expression of genes associated with osteogenesis, such as runt-related transcription factor (Runx2), osterix, osteocalcin and osteopontin [123]. The suppression of Runx2, as one of the classic osteogenic markers and key transcription factors, not only suppresses osteogenesis, but also promotes adipogenesis of MSCs [121]. Moreover, Runx2 is also a target for miR-204/211 to inhibit osteogenesis [124]. In addition to the above, even in the process of a certain differentiation, the levels of the same miRNA can constantly change, which is characteristic of a complex, dynamic differentiation process dependent on various conditions [121]. 

The modulation of miRNA signaling in in vitro or endogenous SC populations as part of a tissue engineering strategy can provide a useful tool for managing tissue regeneration [117]. However, more research is required to determine whether miRNAs and siRNAs can be used to treat a particular disease.

## 6. Conclusions

The technology of mobilization/homing resident SCs for the regeneration of damaged organs/tissues based on endogenous healing mechanisms has become a new concept in regenerative medicine, known as ERM. The accumulated data indicate the possibility of resident quiescent SCs of various tissues to activate the physiological regenerative ability of a tissue. Taking these data into account, it is possible to circumvent the costs and complexity associated with the exogenous regenerative approach, which includes the cultivation of SCs and the creation of functional tissues in vitro by using an endogenous regenerative approach, in which the organism is used as an in vivo bioreactor for tissue regeneration. Since the natural endogenous regenerative process is usually too limited, strategies are being developed for the successful regeneration of many tissues that promote the recruitment of resident SCs to the damaged area. This approach opens a new direction of research focused on the use of the latent regenerative potential of the patient’s own cells, which makes this type of tissue engineering safer, simpler, and more practical and economical than other approaches, while achieving effective and successful results. 

Nevertheless, ERM remains an emerging field of research, and many details concerning the cellular and molecular events that drive the homing of endogenous cells, and their recruitment from tissue-specific niches remain unknown. This concerns both the engineering of biomaterials in the creation of matrices necessary for the proliferation and differentiation of recruited cells, and, consequently, the regeneration of new tissues, and a combination of growth factors, cytokines and other biological signals specific to each tissue. Aside from this, the ability of the organism to regenerate can be limited in patients with an insufficient number of resident cells (e.g., elderly patients) or in tissues with an internally insufficient pool of endogenous SCs and/or low regenerative potential. Even in cases when endogenous SCs alone fail to realize their therapeutic promise, the identification of key regulators involved in SC homing is highly valuable for understanding the native regenerative process and directing future tissue-engineering design. Future expanding opportunities in this field will continue to foster strong collaborative efforts among cell biologists, clinicians, material scientists and engineers.

## Figures and Tables

**Figure 1 ijms-22-13454-f001:**
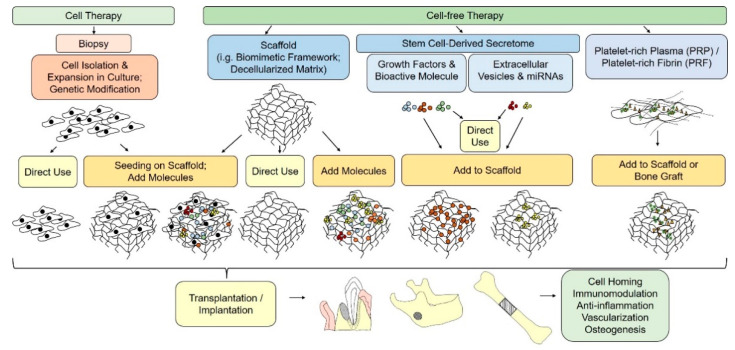
Schematic illustration of cell-based and cell-free paradigms for bone tissue engineering.

**Figure 2 ijms-22-13454-f002:**
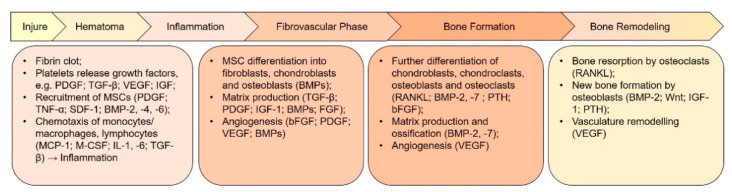
Key growth factors and events involved in the different phases of bone regeneration. The regeneration process can be divided into several phases that overlap each other. Each phase is regulated by many cytokines and growth factors secreted by different cell types. Revascularization and angiogenesis are ongoing through the inflammatory phase until the bone formation phase. BMP, bone morphogenetic protein; FGF, fibroblast growth factor; bFGF, basic FGF; IGF-1, insulin-like growth factor 1; MCP-1, monocyte chemoattractant protein 1; M-CSF, macrophage colony-stimulating factor; OPG, osteoprotegerin; PDGF, platelet-derived growth factor; PTH, parathyroid hormone; RANKL, receptor activator of nuclear factor κB ligand; SDF-1, stromal cell-derived factor 1; TGF-β, transforming growth factor β; TNF-α, tumor necrosis factor α; VEGF, vascular endothelial growth factor.

**Table 1 ijms-22-13454-t001:** ECM composition of connective tissue and bone.

Connective Tissue	Bone
Collagen type I, III, IV, V and X Fibronectin Elastin Fibrillin Laminin Tenascin Nidogen Vitronectin Heparan sulphate Perlecan Versican Biglycan Decorin Fibromodulin Hyaluronan/Hyaluronic acid Syndecan Thrombospondin	Collagen type I, III and V Hydroxyapatite Tricalcium phosphate Osteocalcin Osteopontin/Bone sialoprotein Osteonectin/SPARC Biglycan Decorin Asporin DMP1 MEPE Thrombospondin

## Data Availability

Not applicable.

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
