# Peer review of "Modern Approaches to Acellular Therapy in Bone and Dental Regeneration"

_ijms, 2021, doi:10.3390/ijms222413454_

Round 1
Reviewer 1 Report
The review by Ivanov and colleagues aims to focus on the cell-free therapy approach in bone regenerative medicine. The review is structured in an introduction section, followed by four other sections dedicated to the extracellular matrix, growth factors, secretome and extracellular vesicles, and miRNA. The review topic is interesting and fits with the scope of the International Journal of Molecular Sciences. Overall, the manuscript is easy to read. I have the following observations.
Major points:
- The review fails in covering cell-free therapy approaches for bone regenerative medicine. Indeed, despite the title, the review reports few examples in bone regeneration and many others in fields different from bone regeneration. The review should be modified, focusing better on the bone regeneration field. Please refer to the comments below for more specific details.
- The review seems very superficial in many parts and does not provide any in-depth assessment of the presented literature. It appears as the list narration of many studies lacking any critical assessment of the reported literature. The authors should make a considerable effort to analyzing the selected literature more critically and scientifically. This is especially true when dealing with highly complex products, such as those reported in this review. In this regard, future perspectives regarding their applicability in humans should have empathized.
- Introduction section. The logical flow of the information reported must be improved. After introducing regenerative medicine, the authors focus on stem cells, why they have been successful in tissue regeneration, and their limits that can be overcome by a cell-free approach (lines 25-41). After a digression on wound healing (lines 42-58), the section moves to the concepts of endogenous regenerative technology (ERT) and endogenous regenerative medicine (ERM, lines 58-82) and return to highlight the advantages of the cell-free therapy approach (lines 83-91). Lines 83-91 should be moved and integrated after line 42. I cannot see the meaning of the wound healing part, because as mentioned above, an introduction section most focused on bone medicine is required. The part about ERM/ERT, instead, well introduces the following paragraphs, and thus it is fine where it is reported.
- Extracellular matrix section. Even here, the logical flow has to be improved. I would start with the definition of ECM (lines 110-128) and then move to its employment in ERM (lines 99-109). Lines 132-148 report only limited studies for bone medicine, and they must be integrated. Moreover, as the review should be focused on bone, the reference to other tissues, such as ligaments, dermis, etc., should be removed and replaced with examples specific to bone medicine. At the end of the section, it is fine to conclude by discussing the decellularization methods, but they are not the only limitation to ECM use. Therefore, a less superficial discussion about their limitations and some future perspectives are needed.
- Growth factors and signalling molecules. The logical flow of this section is acceptable; however, it is more focused on dental tissues and wound healing instead of bone. Therefore, more bone examples should be provided. Even here, a less superficial discussion about their limitation and some future perspectives are needed.
- Secretome and extracellular vesicles section. This section is sufficiently concentrated on bone, and its logical flow is fine. Line 280: a reference is needed. Also, when citing the need for scalability and GMP grade products, the authors should consider that manuscripts reporting scalable and GMP-compliant isolation precesses for secretome and EVs have been published. Although they still show limitations, they must be cited.
- Noncoding RNAs section. This section is not focused on bone medicine.
Minor points:
- The review must also be improved by adding figures and tables; this would undoubtedly help its diffusion in the scientific community.
Author Response
We are very grateful to the Reviewer for all critical comments and constructive offers. According to the proposals of the reviewer, we have removed the paragraph on wound healing (lines 42-58) in the Introduction and made many changes to the text of the manuscript. We reworked the extracellular matrix (lines 138-144; 185-187; 190-199; 205-209; 212-214; 222-229; 231-242) and growth factors (249-262; 298-316320-323) sections, adding examples of their use in bone regeneration. In addition, we added a discussion of the problems and limitations of decellularized matrix and growth factors in use in the clinic. In section - noncoding RNA, we reworked the text (lines 468-475;477-484) and added examples of the use of noncoding RNA in bone regeneration. In addition, we have added two figures and one table to illustrate the data discussed. Because bone regeneration is closely associated with craniofacial problems and dentistry, we have corrected the title of our manuscript. All changes in the manuscript are highlighted in green.
Reviewer 2 Report
thank you for this very informative and interesting paper. With this review, the authors create a good summary of the most important approaches in cell-free regenerative therapy in a clear and beginner-friendly way. Molecular relationships are explained in an understandable way and the various aspects of cell-free therapy are discussed holistically with their respective strengths and weaknesses.
On the negative side, however, it is noticeable that long lists of the various cytokines, receptors or secretory vesicles significantly disrupt the flow of reading and may be therefore not completely conceived by the reader. Therefore, the use of overview graphics, for example as mind maps or as tabular overviews, would have a considerable value for the paper and is strongly recommended.
Apart from that, we consider the paper to be a contribution with added value, especially for beginners in the field of cell-free therapy.
Author Response
We wish to thank the Reviewer for the attention to our work. We have added two figures and one table to illustrate the data discussed. Because bone regeneration is closely associated with craniofacial problems and dentistry, we have corrected the title of our manuscript and added a discussion of the problems and limitations of decellularized matrix and growth factors in use in the clinic.
Reviewer 3 Report
According to the title, the authors are focusing their review on “Cell-free Therapy in Regenerative Medicine of Bone: Current Approaches” and the readers are expecting the summary and critical discussion of different modern approaches used for bone healing. The authors discuss ECM, Growth factors and signaling molecules, secretome, extracellular vesicles and non-coding RNAs in regenerative medicine; however, the novelty of the information/special focus/novel point of view is absent.
The text in present form is bringing broad and superficial overview of the different concepts used in regenerative medicine and does not fill criteria of a good review. There is no new angle that has not been covered adequately in the previous reviews; the authors do not crystallized the major achievements in the reviewed field and the main areas of debate.
In addition, the review miss the tables, schematic drawings and figures summarizing findings of recent original research studies, which could support and strengthen the text.
Unfortunately, in the present form the manuscript could not be accepted for publication and should be rejected.
Author Response
We wish to thank the Reviewer for the attention to our work and its opinion. We reworked the manuscript in all sections and added a discussion of the problems and limitations of acellular therapy in the clinic. We have added two figures and one table to illustrate the data discussed. All changes in the manuscript are highlighted in green.
Round 2
Reviewer 1 Report
I am satisfied with the revision provided.
Author Response
We wish to thank the Reviewer for the attention to our work and its opinion.
Reviewer 3 Report
The authors have significantly improved the manuscript, however the extensive editing of English language and style is required.
Author Response

(The authors gave the same response as above.)
